# Single-crosslink microscopy in a biopolymer network dissects local elasticity from molecular fluctuations

Lingxiang Jiang [1], Qingqiao Xie[1], Boyce Tsang[2] & Steve Granick[3,4]

Polymer networks are fundamental from cellular biology to plastics technology but their intrinsic inhomogeneity is masked by the usual ensemble-averaged measurements. Here, we construct direct maps of crosslinks—symbolic depiction of spatially-distributed elements highlighting their physical features and the relationships between them—in an actin network. We selectively label crosslinks with fluorescent markers, track their thermal fluctuations, and characterize the local elasticity and cross-correlations between crosslinks. Such maps display massive heterogeneity, reveal abundant anticorrelations, and may contribute to address how local responses scale up to produce macroscopic elasticity. Single-crosslink microscopy offers a general, microscopic framework to better understand crosslinked molecular networks in undeformed or strained states.

[1] College of Chemistry and Materials Science, Jinan University, 510632 Guangzhou, China. [2] Department of Physics, University of Illinois, Urbana, IL 61801, USA. [3] Center for Soft and Living Matter, Institute for Basic Science (IBS), Ulsan 44919, Republic of Korea. [4] Departments of Chemistry and Physics, UNIST, Ulsan 44919, Republic of Korea. Correspondence and requests for materials should be addressed to L.J. (email: jianglx@jnu.edu.cn) or to S.G. (email: sgranick@ibs.re.kr)

 1

While crystallography is largely successful in describing and modeling crystalline materials with a degree of periodicity such as silicon and steel, a universally successful methodology is still lacking for amorphous or soft materials with intrinsic microscopic disorder and heterogeneity[1–3]. Of special importance are polymer networks, a ubiquitous soft material that fundamentally pervades numerous applications in chemistry, materials science, and biology[4–7]. While classical rheological and scattering measurements in bulk samples extract ensemble-averaged information[8–11], recently developed microrheology shift the paradigms to spatially resolved viscoelasticity[12,13].

The fundamental principle of any rheology is stress–strain dependent; for instance, elasticity of a spring is known per Hooke's law once the applied force and deformation are determined. Microrheology as implemented traditionally pushes this principle to mesoscale by employing colloidal particles as probes[12]. Specifically, probe particles are immersed in a material of interest, they are driven by thermal energy or external forces (magnetic, optical) of known magnitudes to stress the surrounding matrix, their displacements are recorded to infer matrix deformation, and viscoelasticity local to the probes is deduced from the perspective of, for example, the generalized Stokes–Einstein equation[12]. When using probe particles in this way, the method is fundamentally limited to the mesoscale because of the underlying assumption of continuum[12]. The probe particles must possess well-defined sizes, a few times the network mesh size, precluding the use of molecular crosslinks or network segments as probes. In addition, introduction of external particles is often invasive, as they may disturb network architecture by creating local "pockets"[13].

In this paper, we establish a minimal conceptual framework to address the stress–strain (thermal energy and fluctuations) dependence of molecular crosslinks and develop an experimental methodology, referred to as single-crosslink microscopy, to visualize elasticity on the molecular level. We elect to implement the method and verify the model in an in vitro network of actin filaments, because they have been well characterized by macroscale and mesoscale methods, and are highly biologically relevant[14,15]. Prevailing in eukaryotic cells, actin is the main constituent of the cytoskeleton—a network of filaments, crosslinking proteins, and motor proteins—which play a crucial role in cellular deformation, mobility, and division[14,15]. Elasticity of cytoskeletal networks on the molecular level is expected to be intimately related to force generation and transmission in living cells. While complexity of actin–protein regulation is at the heart of the actin network function in a biological context, here we study a minimal actin network as a typical instance of generically occurring, percolated networks. Specifically, we crosslink the actin filaments by the biotin–avidin linkage, label the crosslinks in a specific and noninvasive manner, track their fluctuations by fluorescent microscopy, and translate the fluctuations into local elasticity via the minimal model. We finally map out the local elasticity and cross-correlations between nearby crosslinks in real space, highlighting massive heterogeneity, and anticorrelations.

## Results

**Methodological design and experimental realization.** As outlined schematically in Fig. 1a, single-crosslink microscopy is composed of four steps including labeling, whereby crosslinks are labeled specifically and preferably in a noninvasive way, imaging, whereby videos of the thermally fluctuating network are recorded in real time, tracking, whereby crosslinks' trajectories are extracted from the videos, and mapping, whereby local elasticity and cross-correlations are resolved. We argue that the resulting

microscopic map with fluctuation and elastic information in real space is helpful to understand crosslinked molecular networks, much like the importance of a road map detailing places, roads, and transportation to a traveler.

Although the concept could be in principle extended to other macromolecular networks with crosslinks also labeled appropriately, we focus on in vitro networks of actin filaments crosslinked by biotin–avidin linkage. In these networks with relatively long, semiflexible strands, the length between crosslinks is conveniently larger than in networks of synthetic polymers whose strands are typically shorter and more flexible. A protocol was designed to polymerize the filaments and to fluorescently label the crosslinks by exploiting the fragmentation/fusion nature of actin filaments (Fig. 1b, see Methods section for details)[16,17]. The resulting network is composed of a "sea" of unlabeled filaments and a few labeled ones (ratio ~1000:1), in which green segments with biotin are most likely to be crosslinked to unlabeled filaments (Fig. 1c). The biotin–avidin linkage was chosen for its high binding constant and rigid engagement of two filaments[18], roughly equivalent to an orthogonal, tetrafunctional junction (Fig. 1c, inset). The present labeling is noninvasive and specific to single crosslinks, whereas colloidal particles in microrheology may disturb local network and extract properties averaged over many crosslinks.

In a typical microscopy image (Fig. 1d), the labeled filaments manifest themselves as red backbones and the crosslinks as sparse, green segments, which serve as point sources of fluorescence emission to be tracked in real time with subdiffraction resolution (~20 nm, see Methods section for details). As highlighted in Fig. 1e, the crosslinks fluctuate about their mean positions with trajectory clouds of nontrivial anisotropicity and polydispersity of size and shape. Each trajectory was fitted by a 2D elliptical Gaussian function to identify its center and long and short axes, $a$ and $b$. We normalized ~$10^3$ trajectories (~$10^6$ data points in total) by $b$ and overlapped them to produce an ensemble probability distribution in Fig. 1f (the black mesh). The Gaussian fitting (the rainbow-color surface) matches the data well, giving a significant overall ellipticity $<a/b> = 1.4$.

**Minimal model to correlate cross-link fluctuations and local elasticity.** With full details presented in the Methods section, essential elements of the model are sketched here. Consider a semiflexible filament of contour length $\ell$ and persistence length $\ell_p$ clamped to a fixed orientation and position at one end and free to fluctuate at the other (Fig. 2a). Effective spring constants $k_\perp$ and $k_\parallel$ are respectively parallel and transverse to the clamp orientation with the ratio $k_\parallel/k_\perp = 2\ell_p/\ell$[19,20]. While the transverse response (bending) is of purely mechanical origin and hence enthalpic, the parallel response is dictated by thermally excited undulations and hence entropic. In the stiff limit of $\ell_p/\ell \gg 1$, the highly anisotropic, spring constants constrain the free end to an oblate spheroidal bulb (Fig. 2a, the cyan pancake). These features distinguish semiflexible polymers from their flexible counterparts whose response to forces is isotropic and entropic in origin.

As the junctions in our system are designed to be tetrafunctional if reaction is complete, and roughly orthogonal, we consider two filaments normal to each other, clamped at four ends and joined in the middle with four equal clamp-to-crosslink distances $\ell_c$ (Fig. 2b). Thermal response of the crosslink is determined by a parallel connection of the four individual effective springs with $k_\perp^c = 4k_\perp$ and $k_\parallel^c = 2k_\perp + 2k_\parallel$, such that $k_\parallel^c/k_\perp^c = \left(1 + \ell_p/\ell_c\right)/2$. Given $\ell_p = 16\,\mu m$ and $\ell_c \approx 1\,\mu m$ for the current actin networks, $k_\parallel^c/k_\perp^c \gg 1$ and the cross-link trajectory is a prolate spheroidal bulb (Fig. 2b, the cyan spindle). For each $\ell_p$

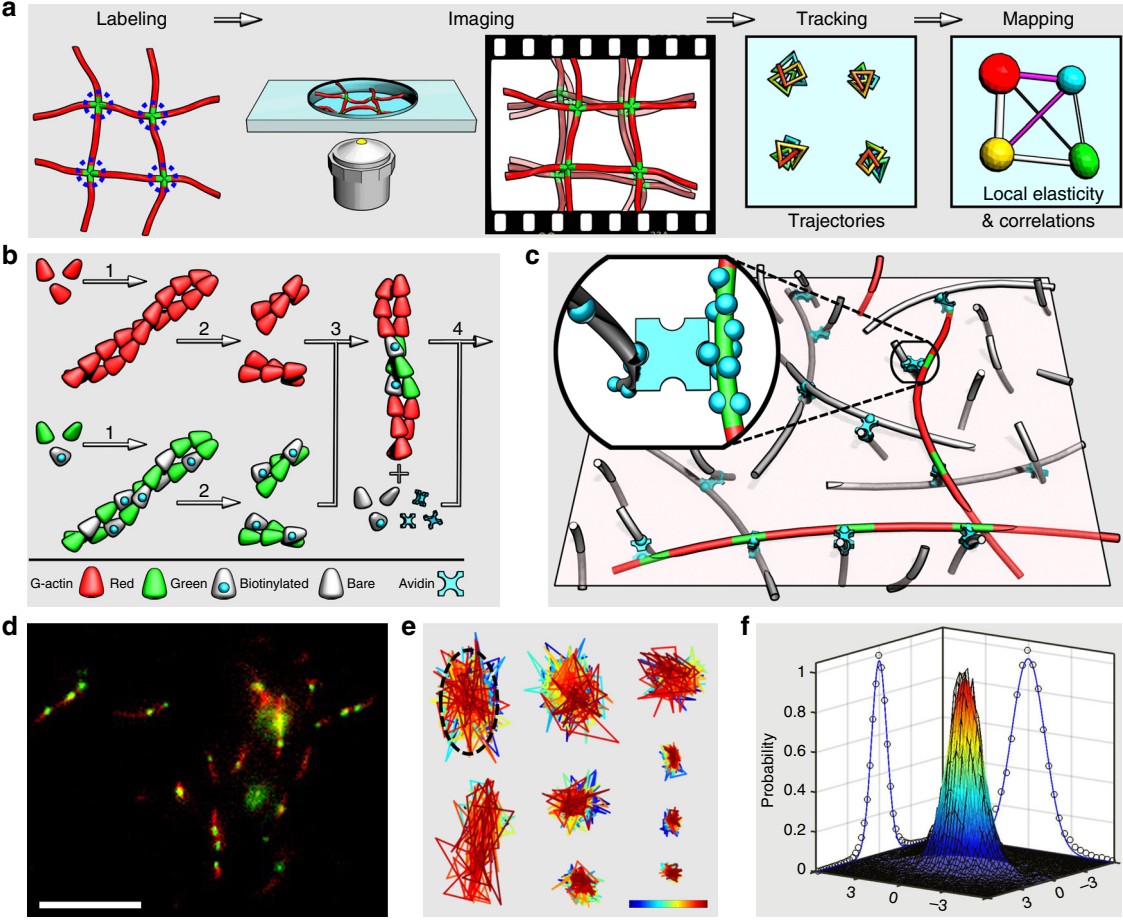

**Fig. 1** Single-crosslink microscopy in actin networks. **a** Four steps of single-cross-link microscopy, see text for details. **b** Four steps to produce the network are (1) pre-polymerization to form red, biotin-free F-actin and green, biotin-loaded F-actin, (2) fragmentation to obtain ~100 nm segments by vigorous shear, (3) fusion to anneal the segments into long filaments with red, biotin-free backbones and sparsely distributed green segments with abundant biotin, and (4) network formation to obtain a cross-linked network of unlabeled and labeled filaments (ratio ~1000:1). **c** Schematic representation of the crosslinked network on the microscope focal plane, showing a few labeled filaments with red contour and crosslinkable, green segments in a background of unlabeled filaments. Inset highlights a biotin–avidin linkage in the orthogonal configuration. **d** Typical microscopy image of labeled filaments. The red contours are generally in the focal plane, and the green segments are traced as time elapses. Scale bar = 10 μm. **e** Color-coded crosslink trajectories highlighting anisotropicity and polydispersity. The color bar from blue to red covers 0–500 s with 0.5 s step. Length of color bar = 400 nm. **f** The ensemble probability distribution (black mesh) against displacements rescaled by $b$ is fitted by a 2D Gaussian peak (rainbow-color surface with color denoting height) with projections plotted on the side walls (circles and curves are data and fit, respectively)

and $\ell_c$, the fluctuation bulb is a specific, 3D Gaussian cloud with widths $\Delta_\perp$ and $\Delta_\parallel$ (Fig. 2c). In practice, we chose to track labeled filaments roughly within the focal plane, so the other filaments are oblique to the plane $(90°-\theta)$. Projection of the 3D cloud on the plane is a 2D elliptical Gaussian distribution of widths $a$ and $b$ and with ellipticity dependent on $\theta$; as an example, simulated data are shown for $\ell_p = 16$ μm, $\ell_c = 1$ μm, and $\theta = 40°$ (Fig. 2d). This prediction of ellipticity is in a qualitative agreement with our observations (Fig. 1e, f).

Applying this concept to actual networks, we consider neighboring crosslinks to be clamps and make a simplification that the four arms are equally long, such that $\ell_c$ denotes local crosslink spacing. It is, however, impractical to directly visualize $\ell_c$ in real space, as the crosslinks are sparsely labeled. We thus take a sidestep to measure projected fluctuations of a target crosslink and to sequentially deduce its $\ell_c$ and local elasticity. A diagram of workflow demonstrates mutual dependences of the key variables (Fig. 2e), including two measurables ($a$ and $b$), three parameters defining the spindle ($\Delta_\perp$, $\Delta_\parallel$, and $\theta$), a central quantity ($\ell_c$), local elasticities parallel and transverse to the filaments ($E_\parallel$ and $E_\perp$), and two input parameters ($\ell_p = 16$ μm for actin

filaments and mesh size $\xi = 0.7$ μm for the current actin concentration of 0.2 mg ml$^{-1}$). Notably, at the heart of this diagram is $\ell_c$ that determines the spindle geometry and local elasticity.

With no adjustable parameter, except $\ell_p = 16$ μm, this minimal model predicts power-law dependence of $\ell_c$, $\Delta_\perp$, and $\Delta_\parallel$ on $b$ (Fig. 2f), with a gap ~1.5 decade between $\ell_c$ and $\Delta_\perp$ and a gap ~0.5 decade between $\Delta_\perp$ and $\Delta_\parallel$. Seeking to test the model, the experimentally measured $a'$ (binned by $b$ values, gray points) are compared with the theoretically predicted $a'$ (averaged over $\theta$, gray line, see Methods section for details). The overall consistency is satisfactory, while the slight systematic overestimation may suggest secondary contributions from unequal $l_c$ or off-orthogonal configurations. Following the arguments in refs. [15,19], we assume a $\ell_c \times \xi \times \xi$ box occupied by the filament between two crosslinks and then proceed to calculate local elasticities (Fig. 2g), where $E_\parallel$ is roughly two orders of magnitudes larger than $E_\perp$.

**Construction of a dynamic map.** Crucial to understanding a network is a map—a symbolic depiction of spatially distributed

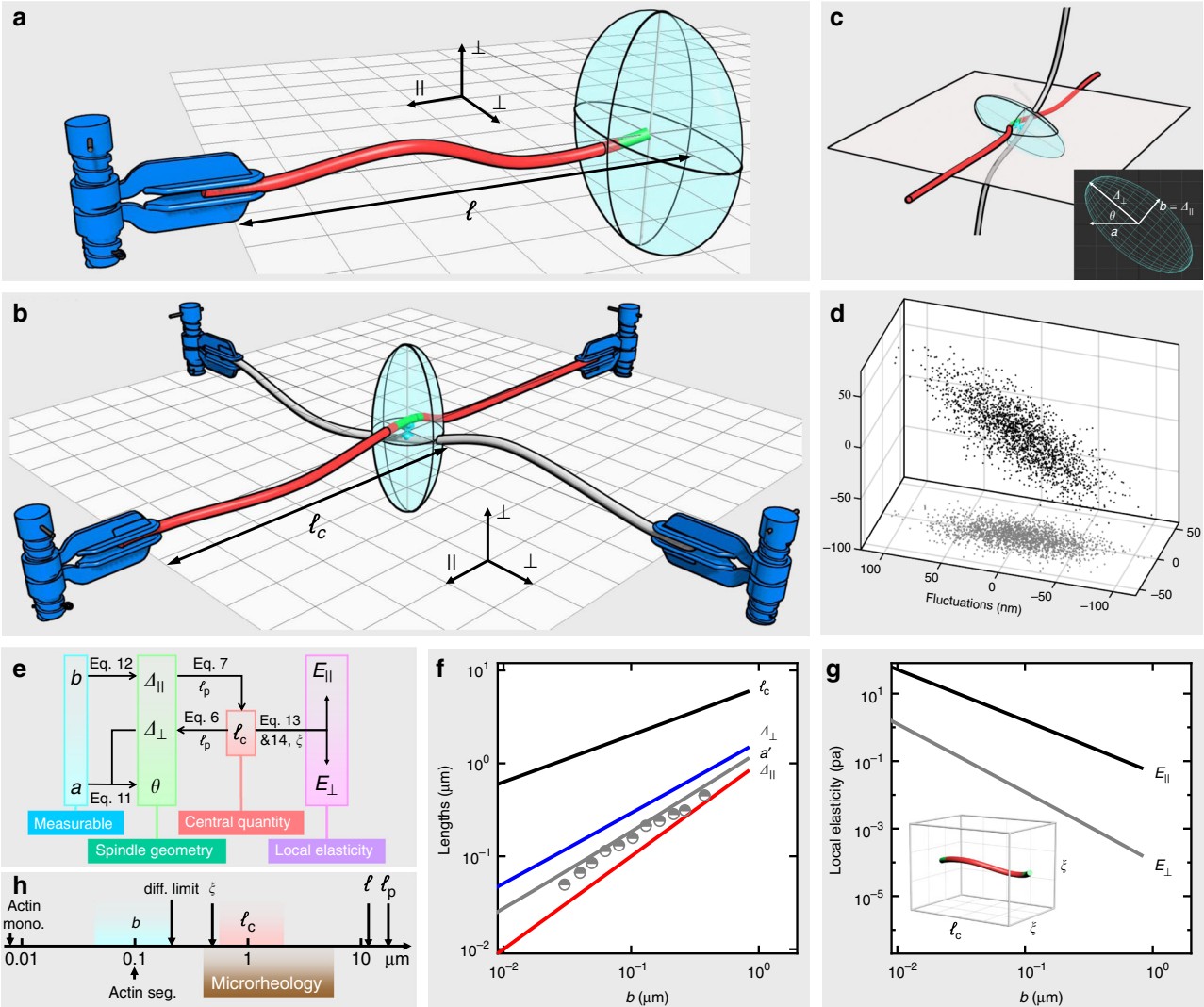

**Fig. 2** Theoretical model for single crosslinks. **a** A filament (contour length $\ell$) is clamped at one end and allowed to fluctuate thermally in a trajectory denoted by the pancake bulb at the other end. **b** Two filaments crosslinked at their middle such that the junction fluctuates thermally in a trajectory denoted by the spindle bulb. Here, $\ell_c$ is the clamp-to-junction distance (or crosslink distance). Parallel and vertical directions are denoted by $\parallel$ and $\perp$. **c** We chose to track the labeled filament that lies within the focal plane; the other filament is oblique to the plane. The 3D spindle is defined by $\Delta_\perp$, $\Delta_\parallel$, and $\theta$. Its projection on the focal plane is a 2D elliptical Gaussian cloud with long and short axes ($a$ and $b$) experimentally measurable. **d** Given $\ell_p = 16\,\mu m$, $\ell_c = 1\,\mu m$, and $\theta = 40°$, a trajectory is simulated. **e** A diagram of workflow showing how to calculate different quantities with two measurables $a$ and $b$. **f** Predicted dependence of different lengths ($\ell_c$, $\Delta_\perp$, $\Delta_\parallel$, and $a'$) on fluctuation magnitude $b$. See methods for the definition of $a'$. The dependence of $a'$ against $b$ shows good agreement between the experimental data (gray points) and predicted line (gray line). **g** Predicted dependence of local elasticity ($E_\parallel$ and $E_\perp$) on fluctuation magnitude $b$. Each filament between two crosslinks is assumed to occupy a $\ell_c \times \xi \times \xi$ box, where $\xi$ denotes mesh size. **h** Relevant length scales. From large to small, they are filament-persistence length $\ell_p = 16\,\mu m$, contour length $\ell$ ~10 μm, microrheology probe size on the order of 1 μm, $\ell_c$ ~0.7–2.5 μm, mesh size $\xi \approx 0.7\,\mu m$, fluctuation magnitude $b$ ~30–200 nm, diffraction limit = 200 nm, length of green actin segment ~100 nm, size of actin monomer = 8 nm

elements highlighting their physical features and the relationships between them. Following this argument, we constructed maps of crosslinks encoded with their fluctuations, local elasticity, and cross-correlations between them. In a typical map (Fig. 3, see Methods section for map construction), a crosslink is represented by a spindle positioned at the trajectory center, size determined by fluctuation magnitude $b$ (enlarged 50 times for clarity), and color coded by local elasticity. Large spatial heterogeneity is obvious for fluctuations and local elasticity with the latter spanning a factor of 10 in this single view. Stiff and soft regions coexist with no apparent correlation between them. How does thermal fluctuation of a given crosslink couple to that of another? Treating here the simplest question of two-point cross-

correlations, we defined the motion correlation $C_{rr}$ between two junctions with a separation $r_0$ in a crosslinked network (Fig. 3, inset and caption, see Methods section for details) and enrich the map with correlation information (lines connecting the spindles). Quite a few negative connections (pink) coexist randomly with the positive (white) and close-to-zero ones (black).

Next, we look at these properties from a statistical perspective. For samples with 0.5, 1.5, and 3% biotin concentrations, higher biotin concentration corresponds to shorter crosslink distance (Fig. 4b), less fluctuation (Fig. 4a), and higher local elasticity (Fig. 4c), as expected on physical grounds. Their high polydispersity narrows with increasing crosslink concentration. For instance, the local elasticity (blue points in the right panel of

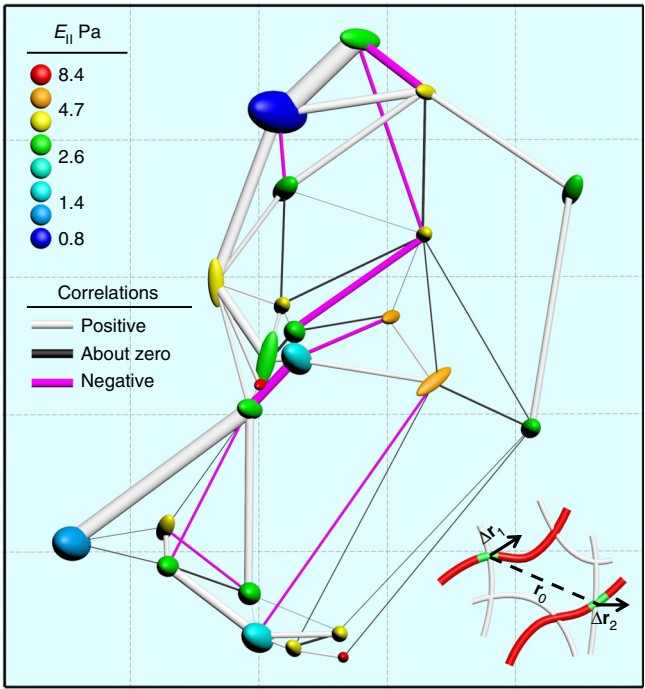

**Fig. 3** A spatial map of fluctuations, local elasticity, and correlations between crosslinks. In this 42 × 45 μm view, the fluctuation bulbs are sized according to $b$ (enlarged 50 times) and color coded by $E_\parallel$. The connecting lines denote cross-correlations ($C_{rr}$) with their widths and color representing magnitudes and signs. The connected crosslinks are not necessarily neighbors, there could be many unlabeled crosslinks in between them. Massive heterogeneity and negative lines of moderate magnitude are noteworthy. The inset shows crosslinks 1 and 2, which are $r_O$ apart and experience displacements of $\Delta\mathbf{r}_1$ and $\Delta\mathbf{r}_2$ during an elapsed time. $C_{rr}$ is defined as the product of the projected displacements (on the $\mathbf{r}_O$ vector) ensemble-averaged over many time steps, $C_{rr} \equiv \langle\Delta r_{1,r}\Delta r_{2,r}\rangle$. Please note that this map was measured and constructed in 2D

Fig. 4a) spans two orders of magnitudes, echoing our observation in the map (Fig. 3). We derive elasticity on mesoscale by dividing a full map into small tiles and averaging elasticities within each tile (Fig. 4d). The mesoscopic elasticity gradually converges to a common mean when the length-scale approaches the full size of our microscope view (82 μm). We expect the convergence to take place at a much smaller length scale, if all the crosslinks (labeled and unlabeled) are counted.

## Discussion

We summarize different length scales relevant to this work in a diagram (Fig. 2h). Known lengths specific to actin filaments include contour length $\ell$ ~10 μm, persistence length $\ell_p = 16$ μm, and monomer size = 8 nm. The mesh size or average distance between filaments $\xi$ can be estimated from actin concentration $c$ in mg ml$^{-1}$ as $\xi \approx 0.3/\sqrt{c}$ in μm, giving $\xi \approx 0.7$ μm for 0.2 mg ml$^{-1}$. Length of green-labeled segments ~100 nm defines the probe length scale of the current single-crosslink method, an order of magnitude smaller than typical for microrheology (~1 μm). The measured fluctuation magnitude $b$ ranges from 30 nm to 200 nm (Fig. 2h, cyan area, and Fig. 4a), corresponding to crosslink distance $\ell_c$ from 0.7 μm to 2.5 μm (Fig. 2h, pink area, and Fig. 4b). Considering each crosslink to consume two biotin sites and assuming all the biotin sites to be crosslinked, we estimate the lower limits of average $\ell_c$ to be 1.5, 0.6, and 0.3, respectively, for biotin

concentrations = 0.5, 1.5, and 3% and a given monomer size = 8 nm. In comparison, the experimental peak $\ell_c$ (Fig. 4b) is 1.8, 1.4, and 1.1 μm, respectively, reasonably larger than the lower limits; the discrepancy increases probably because more biotin sites are not crosslinked at higher biotin concentration.

Existing simulation and theory suggested dominance of parallel local elasticity for a triangular lattice and prevalence of softer vertical modes in other regular architectures[20], and experiments supported the former view when considering a densely crosslinked actin network. Here, we attempt to compare the local and global elasticities. One way to measure the macroscopic plateau modulus $G^0$ is two-point microrheology, whose length scale is defined by the distance between probes rather than the probe size[13]. Although labeled molecular crosslinks were employed as the probes in this particular case, the two-point microrheology is still mesoscopic or macroscopic because the probe distance is larger than 1 μm and it must be ensemble averaged over multiple probe pairs. The measured $G^0 = 0.2$ Pa for biotin concentration = 1.5% sits in the middle of the averaged local elasticities $\overline{E_\parallel} = 4$ Pa and $\overline{E_\perp} = 0.04$ Pa. It is tempting to deduce a mixed contribution from parallel and vertical modes to the macroscopic modulus. However, it is unfeasible to extrapolate, at this stage, from the local to the global because network geometry, filament orientation, and network affinity are not exactly known.

While we focus on bending and undulation modes of actin filaments, there are reports that in actin networks, twisting and bend-twist coupling modes[21] contribute as a result of the double-stranded helical organization of actin subunits. The twisting and coupling persistence lengths are, respectively, 1.4 and 0.4 μm, relevant to $\ell_c$ ~1 μm in this paper. Present understanding of their contributions seems to be that these modes are prominent on a time scale of μs to ms and most relevant to small filament deformations like subunit incorporation (~8 nm)[21]. In the current experiments, the frame rate is slow (0.5 s) and deformation is larger (~100 nm). Thus, it is likely that these subtle modes do not dominate the crosslink fluctuations analyzed here.

Next, we consider the cross-correlations between crosslinks. Hydrodynamic arguments for a continuum anticipates $C_{rr}r_0$ to be constant regardless of separation[13,17], whereas our data is massively scattered (Fig. 4e) with negative points highlighted by the pink background. The probability distribution of $C_{rr}r_0$ on a semilog scale (Fig. 4f) is a peculiar triangle peaked close to zero, decaying exponentially, and slightly biased to the positive. The distribution broadens as the crosslink concentration decreases. These systematic anticorrelations (negative $C_{rr}$) are astonishing considering their abundance and statistical regularity.

In the literature, anticorrelations were previously observed in an active system of actin networks loaded with myosin, a motor protein, and were attributed to contractile motions powered by myosin;[22] also, negative local elasticity was found to coexist with positive local elasticity in simulation of a quenched polymeric glass[23]. In the present case, we argue that the following may contribute significantly. The key point is that filament bending shortens or lengthens the distance between two crosslinks, causing possible anticorrelations (Supplementary Fig. 1a). Considering four crosslinks on rhombus corners (Supplementary Fig. 1b), the diagonal motion of one crosslink would cause two neighboring crosslinks to move against or away from each other. We notice that crosslinked filaments are generally short (~5 to 10 μm) and occasionally in a kinked conformation (probability ~5%, Supplementary Fig. 1c), whereas uncrosslinked filaments are long (~10–20 μm) and smooth[17]. We suspect that the crosslinking kinetics may shorten the filaments and lock them

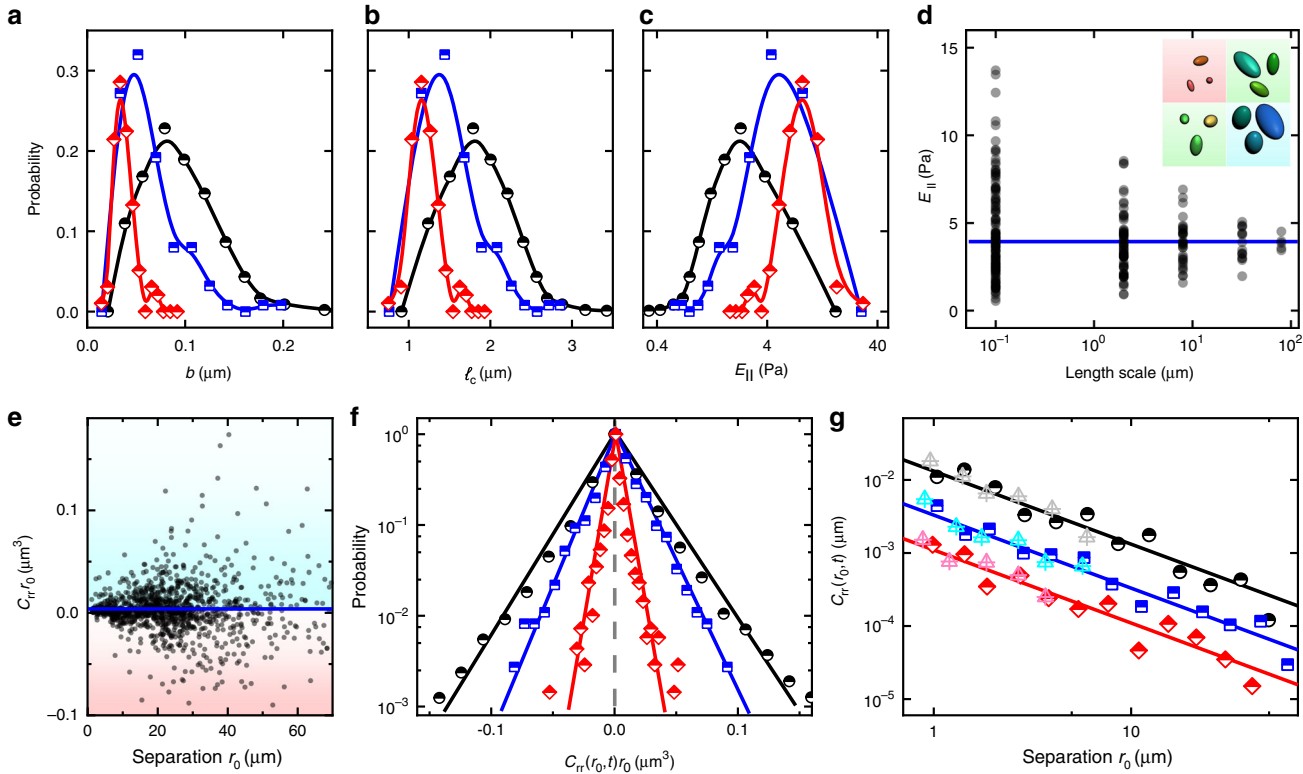

**Fig. 4** Statistics of single-crosslink fluctuations, local elasticity and two-crosslink cross-correlations. **a–c** Probability distributions of crosslink fluctuation (**b**), local crosslink distance ($l_c$), and local elasticity ($E_\parallel$) and their dependence on crosslink concentration. Lines are guides to the eye. **d** Mesoscopic elasticity averaged over a tile. The points converge to a common mean (blue line) with larger tile size. **e** Distribution of $C_{rr}r_0$ against $r_0$. Scatter shows individual crosslink pairs and blue line shows their average. The positive and negative region is shaded by cyan and pink, respectively. **f** Probability distribution of $C_{rr}r_0$ displays a peculiar unsymmetrical, triangular shape. Points are experimental data and lines are exponential fits. **g** Dependence of $C_{rr}$ on separation for intra-filament (light points) and inter-filament (heavy points) crosslinks. The lines are the classic hydrodynamic continuum prediction $1/r_0$. For comparison, see the data for an entangled, uncrosslinked network in Supplementary Fig. 2. In **a–c**, **f**, and **g**, biotin concentrations are 0.5, 1.5, and 3% for black, blue, and red points, respectively. In **d**, **e**, biotin concentration is 1.5%

into kinked configurations and nonequilibrium local states[24,25], contributing to the observed anticorrelations.

Ensemble-averaged analysis of cross-correlations reveals consistency with the continuum prediction for homogeneous materials ($C_{rr} \propto 1/r_0$) over the entire measurable range from 1 μm to 50 μm (Fig. 4g). Despite enormous inhomogeneity and polydispersity on the single-crosslink and two-crosslink levels, these networks can be treated, to our surprise, as a viscoelastic continuum down to the length scale of one crosslink separation on the ensemble-averaged level. Notably, correlations between two crosslinks on the same filament (light points) and on different filaments (heavy points) collapse onto a single line, in great contrast to uncrosslinked actin systems (Supplementary Fig. 2)[17]. By this measure, the filaments lose their "identity" or individuality and become a part of the collective, united network.

Finally, we conclude with a few future prospects for single-crosslink microscopy. The minimal resolvable distance between two crosslinks is limited by optical resolution, ~200 nm in the current setup, while the minimal detectable fluctuation of the crosslinks is limited by the stability and noise of the optical setup, ~20 nm in the current setup. In principle, the former resolution could be improved to ~40 nm using super-resolution microscopy, and the latter to ~10 nm if the system stability and noise were further improved. We presently limit ourselves to the long-time limit with a slow frame rate of 2 Hz, but modern cameras with ms time resolution would enable one to extend this methodology to

construct time-varying maps to, for example, quantify time-evolution of the fluctuation bulbs.

Although the biotin–avidin linkage is rigid and permanent, many biological crosslinkers are flexible and dynamic, rendering the resulting networks even more complex[24]. Previous literature has demonstrated the significant impact on network mechanics of crosslink flexibility and compliance;[24] filament contour length also plays an important role in determining elasticity in such cases[26], but these complexities are beyond the scope of the present study, which has taken the approach of seeking to minimize complexity to the extent possible, while still addressing the network problem. Likewise, a key feature common to many biological networks is strain-stiffening behavior[27] not considered in the current system under equilibrium. We anticipate that extension of actin filaments, using a rheometer for example, or internal contraction of actin filaments, by myosin for example, should both effectively stiffening the filaments and reduce their thermal fluctuations. We also expect the methodology introduced here to be potentially extended to synthetic, flexible polymer systems[9,10,28,29] using fluorescent-labeled crosslinks such as quantum dots and single-fluorophores, especially by capitalizing on recent advances in synthetic means to produce tailor-made networks[30–32].

## Methods
**Materials**. Unlabeled G-actin (rabbit skeletal muscle) and biotinylated G-actin (rabbit skeletal muscle) were purchased from Cytoskeleton Inc. Alexa-568

labeled G-actin (rabbit skeletal muscle) was purchased from Invitrogen. Alexa-647 labeled G-actin (rabbit skeletal muscle) was a kind gift from Prof. William Brieher (University of Illinois). Phalloidin (Amanita phalloides) and streptavidin was purchased from Sigma–Aldrich. All the other chemicals were from Sigma–Aldrich in analytical purity. Water was deionized (18.2 MΩ cm). G-actin was reconstituted in fresh G-buffer (5 mM Tris [Tris(hydroxymethyl)amino-methane] at pH 8.0, supplemented with 0.2 mM CaCl$_2$, 1 mM ATP, and 0.2 mM DTT and 0.01% NaN$_3$) at 4 °C and used within 7 days of reconstitution. G-actin was polymerized into F-actin by the addition of salt (100 mM KCl, 2 mM MgCl$_2$) at room temperature for 1 h.

**Labeling and polymerization.** A four-step protocol was implemented. Pre-polymerization—Alexa-647 (red) G-actin was polymerized into red F-actin, while Alexa-568 (green) G-actin and biotinylated G-actin (molar ratio 2:1) were polymerized into green F-actin. Fragmentation—the red and green F-actin filaments were fragmented into ~100 nm segments by vigorous shear (repeatedly passing them through a 26-gauge syringe needle). Fusion—a mixture of red and green segments (molar ratio 9:1) was incubated 12 h, in the presence of phalloidin to prevent treadmilling and depolymerization, for the segments to fuse into long filaments with red, biotin-free backbones and sparsely distributed green segments with abundant biotin. Network formation—a trace amount (unlabeled-to-labeled ratio = 1000:1) of labeled filaments were mixed with bare G-actin, biotinylated G-actin, streptavidin, and ~30 mM vitamin C (for anti-photobleaching). The mixture was deposited between a glass slide and a cover slip with a spacer (120 μm) to develop a fully crosslinked network ready to be observed by a microscope. Three kinds of samples were inspected in this work with an identical actin concentration (0.2 mg ml$^{-1}$, weakly entangled), an identical biotin-streptavidin ratio (2:1), and different biotinylated-to-bare actin ratios (0.5, 1.5, and 3%).

**Two-color imaging and segmental tracking.** A two-color imaging system was built on an epifluorescence microscope (Zeiss Observer. Z1), where the signal was split to two EMCCD cameras (Andor iXon) by a Y-junction splitter (MAG Biosystems, DC2). We visualized the motion of labeled segments and backbones of F-actin in this home-built system with a ×100 oil objective and focused deep into the sample (>30 μm) to avoid potential wall effects. The field of view was 82 by 82 μm and depth of focus 1.5 μm. Video images were collected typically at 2 fps for 500 s, which were then analyzed by Matlab codes written in-house. The time range was mainly limited by photo-bleaching of labeled actin. The center of each segment was located in each frame to give trajectories with ~20 nm precision. About 150 movies in total were taken. The blurred green points for example in Fig. 1d are out of focus and are filtered out by the Matlab codes.

**Minimal model for crosslink fluctuations.** In the situation of Fig. 2a, the responses of the free end to thermal forces can be characterized by effective spring constants respectively transverse and parallel to the clamp[19,20],

$$k_\perp = 3\kappa/\ell^3 \qquad (1)$$

$$k_\parallel = 6\kappa^2/(k_B T \ell^4), \qquad (2)$$

where $\kappa$ is related to persistence length $\ell_p$ by

$$\ell_p = \kappa/(k_B T). \qquad (3)$$

For the situation in Fig. 2b, where the thermal response of the junction point is determined by the parallel connection of the four individual effective springs, we have

$$k_\perp^c = 4k_\perp = 12\kappa/\ell_c^3 \qquad (4)$$

$$k_\parallel^c = 2k_\perp + 2k_\parallel = 6\kappa/\ell_c^3 + 12\kappa^2/(k_B T \ell_c^4), \qquad (5)$$

where $\ell_c$ is the clamp-to-junction distance, $k_\perp^c$ and $k_\parallel^c$ effective spring constants of the junction in the directions vertical and parallel to the filament plane. By equating the thermal energy to the spring potential $k_B T = \frac{1}{2} k_\perp^c \Delta_\perp^2 = \frac{1}{2} k_\parallel^c \Delta_\parallel^2$, one gets

$$\Delta_\perp^2 = k_B T/(6\kappa/\ell_c^3) = \ell_c^3/6\ell_p \qquad (6)$$

$$\Delta_\parallel^2 = k_B T/[3\kappa/\ell_c^3 + 6\kappa^2/(k_B T \ell_c^4)] = 1/(3\ell_p/\ell_c^3 + 6\ell_p^2/\ell_c^4), \qquad (7)$$

where $\Delta_\perp$ and $\Delta_\parallel$ are the thermal fluctuation magnitudes vertical and parallel to the filament plane. In the limit of $\ell_p/\ell_c \gg 1$, Eq. 7 reduces to

$$\Delta_\parallel^2 = \ell_c^4/6\ell_p^2 \qquad (8)$$

such that $\Delta_\perp \gg \Delta_\parallel$ and the fluctuation bulb is a prolate spheroid (Fig. 2b).

In the experiments, what we can measure is the 2D projection of the 3D fluctuation bulb on the focal plane (Fig. 2c). We chose to track labeled filaments roughly within the focal plane, so the other filaments are oblique to the plane (90° −θ). The projected distribution can be fitted by two Gaussian functions orthogonal

to each other,

$$f_x \propto \exp\left(-\frac{r_x^2}{2a^2}\right) \text{ in the long axis} \qquad (9)$$

$$f_y \propto \exp\left(-\frac{r_y^2}{2b^2}\right) \text{ in the short axis,} \qquad (10)$$

where $r_x$ and $r_y$ are the displacements. Fitting the trajectories with Gaussian functions, we can obtain $a$ and $b$, the fluctuation magnitudes in long and short axes. Recalling the inset of Fig. 2c, they are related to $\Delta_\perp$ and $\Delta_\parallel$ by

$$a = \Delta_\perp \cos\theta \qquad (11)$$

$$b = \Delta_\parallel. \qquad (12)$$

Now that we have related the key parameters to the two measurables (Fig. 2e), we proceed to calculate their dependence on $b$ with the input parameter $\ell_p = 16$ μm for actin filaments (Fig. 2f). Rearrangement of Eqs. 7 and 12 gives $b^2 = 1/(3\ell_p/\ell_c^3 + 6\ell_p^2/\ell_c^4)$, which is numerically solved to predict the dependence of $\ell_c$ on $b$ (black line). This dependence, along with Eq. 6, is further used to calculate the dependence of $\Delta_\perp$ on $b$ (blue line). Dependence of $\Delta_\parallel$ on $b$ is given by Eq. 12 (red line). In our model, 3D spindles with a given $\Delta_\parallel$ take random orientations $\theta$ so we define an averaged quantity $a'$ independent of a particular $\theta$, given by $a' = \langle a \rangle_\theta = \langle \Delta_\perp \cos\theta \rangle_\theta$. Dependence of $a'$ on $b$ is also calculated numerically (gray line). Experimentally, 2D trajectories of similar $b$ values can vary significantly in $a$ because their 3D orientations ($\theta$) are random. We thus group all the measured trajectories into subsets according to their $b$ values and obtain an averaged $a$ value for each subset ($a'$, gray points). Reasonable consistency between the measured and predicted $a'$ values (gray points and gray line) therefore supports our minimal model.

**Calculation of local elasticity.** Following the arguments in ref. [11], we consider a filament between crosslinks of a length $\ell_c$ and assume that it occupies an $\ell_c \times \xi \times \xi$ box in space (Fig. 2g). For deformation transverse to the filament ($\Delta_\perp$, bending), the stress is $k_\perp \Delta_\perp/(\xi\ell_c)$ and the strain $\Delta_\perp/\xi$. Transverse elasticity is given by stress over strain, leading to

$$E_\perp = k_\perp/\ell_c = 3\kappa/\ell_c^4 \qquad (13)$$

For deformation parallel to the filament ($\Delta_\parallel$), the stress is $k_\parallel \Delta_\parallel/\xi^2$ and the strain $\Delta_\parallel/\ell_c$, so parallel elasticity is

$$E_\parallel = k_\parallel \ell_c/\xi^2 = 6\kappa^2/(k_B T \xi^2 \ell_c^3) \qquad (14)$$

**Map construction and cross-correlations.** Extracted from a given video, individual crosslink trajectories were fitted by Gaussian functions to give their centers, $a$, and $b$. We further determined their spindle geometry and local elasticity following the workflow (Fig. 2e). A map was constructed with spindles positioned at the trajectory center, sized by $b$, oriented by $\theta$, and color coded by local elasticity (Fig. 3). Please note that a specific $a/b$ corresponds to two equivalent $\theta$, so we randomly assigned one to the spindle.

The space-time relative displacement correlation function, $C_{rr}(r_0;t)$, is defined as the correlated displacements of two tagged crosslinks as a function of their average separation $r_0$ and elapsed time $t$ during which thermal motion causes displacement amplitudes that are modest relative to $r_0$,

$$C_{rr}(r_0;t) \equiv \langle \Delta r_{1,r}(t) \Delta r_{2,r}(t) \rangle_{r_0} \qquad (15)$$

where $\Delta r_\alpha(t) = \Delta R_\alpha(t) - \Delta R_\alpha(0)$ is the vector displacement of segment $\alpha$ in a time $t$ relative to its initial position, and the subscript r corresponds to projection along the initial separation vector of the two tagged segments. For the connections in the map (Figs. 3 and 4e), the ensemble average $\langle \ldots \rangle_{r_0}$ is performed for a particular pair of junctions with $t = 0.5$ s. For statistics (Fig. 4f), it is performed for all pairs with a separation $r_0$ and $t = 0.5$ s. To do the two-point microrheology calculation, $t$ covers 0.5–50 s. This definition of $C_{rr}$ is essentially the same as that in two-particle microrheology[13]. In our case, $C_{rr}$ is further divided into intra-filament pairs and inter-filament pairs (Fig. 4g and Supplementary Fig. 2)[17]. This division is critical because they could be governed by two entirely different physical mechanisms (Supplementary Fig. 2). In practice, while it is true that our experiments were analyzed as a 2D cross-section through the 3D sample, differences of vertical position are expected to be of secondary influence because the separation between crosslinks greatly exceeded the focal plane thickness ~0.5 μm.

## Data availability
The authors declare that all data supporting the findings of this study are available within the paper and its supplementary information files.

## Code availability
All the Matlab code used to track crosslinks and to calculate local elasticity and correlations is available from the corresponding authors upon request.

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

## Acknowledgements

L.J. acknowledges support by National Natural Science Foundation of China (No. 21773092), Guangdong Natural Science Funds for Distinguished Young Scholar (No. 2018B030306011), and Fundamental Research Funds for the Central Universities (No. 21617320); S.G. acknowledges support by the Institute for Basic Science, project code IBS-R020-D1.

## Author contributions

L.J. and S.G. conceived the experiment and wrote the paper; L.J., Q.X., and T.B. performed the experiments and analyzed the data. All the authors contributed to discussing the results.

## Additional information

**Competing interests:** The authors declare no competing interests.

