## [Peer Review File · Nature Communications]

Reviewers' comments:

Reviewer #1 (Remarks to the Author):

In this manuscript, the authors investigated the effects of microscopic behaviors of crosslinks on the mechanical properties of a confined network based on actin filaments. The proposed microscopy method points to the single crosslink that specifically represents biotin-avidin linkage in an orthogonal configuration, aiming for bridging the gap between dynamics of transient crosslinkers and macroscopic cellular elasticity. The method is limited to studying the elastic behavior of weakly crosslinked actin networks subjected to substantial thermal fluctuations, which are the major driving forces for a local mechanical transition in such scenarios. Using noninvasive labeling and combining it with epifluorescence microscopy, local trajectories of fluctuating crosslinks showed anisotropy and differences in heterogeneity indices regarding the size and shape. To rationalize the observation, a minimal model is developed by constructing a filament-crosslink configuration similar to that in vitro with high binding affinity and rigidity as connecting junctions. Quantitative analysis focuses on the spatial map of local fluctuations and correlations between neighboring crosslinks, and it reveals that the network in such configuration behaves as viscoelastic continuum down to the length scale of one crosslink separation distance. Local deformation is further extrapolated to give quantitative information about changes in local elasticity, and it confirms that the parallel undulation dominates over vertical mode, highlighting importance of entropic behaviors over enthalpy.

Although the analysis showed strong agreement with previous literature, I did not find substantial novelty in the method warranting publication in Nature Communication. Extensive improvements and justification should be made on the extrapolation of local elasticity. Results from the model seem scientifically sound, but it also requires further validation to improve its versatility given the limited spatiotemporal scale and dynamic state. Therefore, I doubt the impact of the work in its current form is sufficient.

< Major Comments >

- The title seems a bit misleading as the focus of the work is the proof of concept of the minimal model the authors developed (since the calculation of elasticity is based on it). The main body of results is based on microscopy. However, I see enough novelty in neither the design of experiments nor the epifluorescence microscopy modified or improved to enable automated tracking/monitoring of weakly crosslinked networks. It is quite convincing that optical approaches are suitable for tracking thermal motions of individual particles. But, in the mechanical perspective, it would be much more efficient to use an AFM-based system or an optical tweezer system to directly quantify the deformation level and time evolutions of forces and velocities of single proteins or particles. The authors should address the novelty and advantages of the method and analysis in an experimental standpoint more clearly. And, I think it might be better to rephrase the title as, for instance, "Mechanical role of single-crosslink microscopic behavior in actin network".

- The authors are assertive with the ability of the microscopy method to characterize local actin mechanics based on a single cross-link. Therefore, this can be very different from the mesoscale properties averaged over a network with many crosslinking proteins. While I would give an enough credit to the theoretical model with the single-crosslink configuration, it seems that the most important quantifications, including ensemble-averaged analysis of cross-correlation among different crosslinks as well as the network elasticity from a statistical perspective, rely on values averaged over the whole actin network. What is the critical/threshold length-scale that differentiates the measured elasticity in this study compared to other mesoscopic properties?

- In the Introduction section, the authors should briefly mention biological cytoskeletal networks that this biomimetic system tries to represent, for strengthening the significance of study. Since the active networks of the actin cytoskeleton undergo a rapid mechanical transition, it would be better if the authors can illustrate the specific role of such a system and the underlying local crosslink mechanism in a biological context.

- Figure 1 does a good job for summarizing the fundamental steps and workflow of the presented method. If Figure 1B specifically introduces the double-stranded helical structure of F-actins, it should also be considered in the discussion or model assumptions. Since it has been hypothesized that the elastic free energy of F-actin at small deformations is heavily affected by the twisting and bending motions due to its intrinsic structure, which emphasizes the significance of the transverse response as opposed to the entropic behaviors giving rise to the parallel response in the study.

- For estimation of the local elasticity, the authors assume cubic geometry for adjacent filaments. This seems problematic as there is no justification or reference from literature for this quantification. If the volume of a single F-actin is calculated in 3D, the deformation Δ should be calculated in 3D for consistency in terms of dimension. Does the deformation have specific directionality in a 3D vector form, or is it assumed to be simply 2D? Given the a relatively small depth of focus in this system, I would assume all measurements are 2D or pseudo-3D at most.

- It is interesting to observe some unexpected anti-correlations from the spatial map. It seems difficult to draw any conclusion on its origins as the authors have mentioned, but I noticed that a negative correlation (Figure 3) only occurs between adjacent crosslinks with relatively different E_{parallel} , which might be worth discussing about.

- It seems counterintuitive that the network of interest in this study, which is only experiencing subtle thermal fluctuation, would result in a state far from equilibrium. As the authors have mentioned, potential buckling might result from crosslinks. However, statistics do not show bending-dominated responses. Therefore, it is possible that twisting might play a role.

- The authors should explain the basis of the proposed configuration of the actin-crosslink network in Figure 3. Are the filaments equally spaced by crosslinks? What is the average length assumed and measured in this work?

- If the spatial heterogeneity in the local fluctuation, elasticity, and correlation is extreme, the authors should provide quantitative maps in addition to the qualitative spatial map. The probability distribution in Figure 4 is beneficial but does not provide details for finding scientific or biological relevance. Is the heterogeneity in local elasticity dependent on cell geometry or network mechanical percolation?

- In similar actin networks, filament length has been shown to be a crucial player in regulating elasticity. Although it might be difficult to tune the filament length in the proposed model, the results might be highly dependent on the filament length in addition to the crosslink concentration. It is also possible that increasing crosslink concentration alters the filament segment length causing kinks, as the author mentioned. It would be clearer if the authors clarify the length scales of filaments both in vitro and in silico in this study.

- The following study also suggested that linear elasticity of weakly crosslinked network is not dominated by crosslinks if the network can be spatially or temporally tuned by other parameters. If so, how sensitive would the proposed network model be as a response to other cytoskeletal-related properties?

“Biophys J. 2010 Aug 9; 99(4): 1091–1100, doi: 10.1016/j.bpj.2010.06.025”

- Figure 4E and Supplementary Figure 3 show distinct behaviors between crosslinked and uncrosslinked actin networks. The author explained well the nonlinearity in the uncrosslinked networks. Uncrosslinked inter-filament networks might be softened more easily. The authors attributed this to dominance of structural forces in uncrosslinked actin systems. In crosslinked networks, I wonder if stiffening of crosslinks would affect network elasticity in this presented system.

- If persistence length is fixed, bending should not contribute too much to the observed behaviors at the length-scales in this study. However, if the authors would extend the methodology to incorporate external forces, how would myosin motors be configured? How would the length-scale be manipulated?

< Minor Comments >

- The predicted line and experimental data show some consistency in Figure 2C. However, it is also noticeable that there is always a certain offset between the two sets. This should be justified.

- In Figure 4 caption, F in "A, B, D, and F" should be E.

- In Figure 1E, it is somewhat difficult to differentiate the trajectories based on the current time scale color bar, and since the elliptical fitting is only drawn for the first cloud of trajectory, it seems unnecessary to include all samples since the majority of them seem inconsistent with the overall ellipticity.

Reviewer #2 (Remarks to the Author):

In this work, the authors developed single-crosslink microscopy to track thermal fluctuations, local elasticity and cross-correlations between nearby crosslinks of actin network. The resulting microscopic map with fluctuation and elastic information in real space is very helpful to understand crosslinked molecular networks. More importantly, the proposed methodology can be extended to construct time-varying maps to measure network dynamics and can be readily used to analyze network properties under external forces.

In my view, this is an amazing work. The presented method bridges the scales from sub-micrometers to centimeters and higher, which are tremendously challenging for the existing microscopy methods. In addition, such method is able to characterize soft materials system with microscopic disorder and heterogeneity, which cannot be studied by recent microrheology method. Therefore, I highly recommend the publication of this work in Nature Communications. One minor comment: What is the special resolution for this single-crosslink microscopy?

Reviewer #3 (Remarks to the Author):

(anonymous report)

The authors report on a clever and potentially very interesting method for visualizing the positions and dynamics of individual crosslinks in a biopolymer network. They look at actin networks, in which the crosslinks come from avidin-biotin linkages, a small fraction of which are fluorescence labeled. From movies of the thermally induced motion, the authors were able to gather statistics at the microscopic scale, which may provide a new kind of insight into statistical mechanics and elasticity of these materials. The idea is an excellent one and the method could be quite useful in future studies. In its present form, however, the manuscript is ambiguous on several key technical and conceptual points. The authors need to clarify in several cases to explain their methods and calculations at an appropriate level and to justify their conclusions. Substantial revisions are needed and it may be that the manuscript has to be significantly lengthened to provide adequate explanation.

Here are my specific comments.

1) The Methods section says that a "trace amount of labeled filaments were mixed with bare G-actin, biotinylated actin..." What does "trace" mean? This is important information because it sets the fraction of crosslinks that are visible by fluorescence. Is it 1 out of 100? 1 out of 10,000? Fig 2C shows an equal number of visible and invisible crosslinks. Is that approximately accurate? The authors wrote that their crosslinks usually have the labeled filament in the plane (p9 L207). Since there would appear to be no reason why the labeled filaments should have a special orientation, does this mean that there is a measurement bias, in which filaments normal to the plane or not detected? This point needs to be explained.

Because only a few of the crosslinks and filaments are visible, it was not clear how the authors know reliably which of the visible crosslinks are connected by filaments and which are not. This was part of the analysis reported near the top of page 6 (L127). It was also reported in Fig S3 without sufficient background of the analysis. The authors' conclusion is that connected and non-connected crosslinks appear the same, but for very basic statistical reasons it seems much more likely to me that all of the *visible* crosslinks were connected by the much larger number of invisible filaments. This would explain why there is no significant difference among them. This part of the analysis needs to be explained in much more detail or perhaps reconsidered. Still following the same question, how do the authors know that the distance between two visible crosslinks is the distance between actual neighboring crosslinks? Figure 2 shows the model, and the text discussion implies that the clamps are representing the nearby crosslinks. What if these clamps are not visible?

2) How do the authors know the 3D orientations and sizes of the ellipsoids shown in Fig 3? The text reports only 2D measurements, which are projections of the 3D motions onto the image plane.

3) Fig 3: are these crosslinks all in the same image plane? From my understanding of the procedure, the correlations can only be measured for pairs of crosslinks that are in the same plane. However, the illustration conveys a sense of perspective, as if the correlations were measured for crosslinks that were in different planes. How is that done? Also, what is the scale for this image?

4) What is the equation that represents the theory curve of Fig 2C? It seems to me that this result depends on the ratio of L_p to L_c . What values were chosen? Also, from the measured values of b and a , I very roughly estimated the ratio of L_c/L_p and found a value of L_c that was on the order of a few microns. This seems much too large for L_c if it is to represent the distance between actual (not just visible) crosslinks. I may have made an error, in which case my point is that this needs to be explained in the text.

Because of these questions, I did not find the discussion of the "local elasticity" to be convincing. What values of L_c are inferred, and how do these compare to what one might expect from the known concentration of crosslinkers? How does it compare to the distance between visible crosslinks, which sets an upper limit?

5) Page 5, L113, negative values of C_{rr} (anticorrelations): Would the following scenario explain the anticorrelation: suppose the crosslinks were connected by a single filament, and suppose that thermal excitations cause this filament to bend, something like a guitar string being plucked. In such a case, the two end points would move toward one another. Does this count as a positive or negative correlation? The other sign of correlation would correspond to the two imaged spots tending to move in the same direction, which could be from long-wavelength modes of vibration. In either case, why is this surprising, and why would such modes be "not expected for equilibrium systems"? This comment needs to be explained.

6) This is a minor comment, but I think the "results" section should be called "results and discussion" and the final section should be called something else.

7) Page 3, L69: The phrase "thermal energy acts on the free end" should be removed. It's not clear what it means for energy to "act" and in any case, there are thermal fluctuations everywhere along the filament, not just at the end.

Response to Reviewer # 1

We thank the reviewer for the extensive comments. The critical issues and insightful suggestions have allowed us to make major revisions to improve the manuscript. We have done our best, in the revised manuscript, to address the reviewer's concerns.

1. Concerns about novelty.

“But, in the mechanical perspective, it would be much more efficient to use an AFM-based system or an optical tweezer system to directly quantify the deformation level and time evolutions of forces and velocities of single proteins or particles. The authors should address the novelty and advantages of the method and analysis in an experimental standpoint more clearly.” “Results from the model seem scientifically sound, but it also requires further validation to improve its versatility given the limited spatiotemporal scale and dynamic state. Therefore, I doubt the impact of the work in its current form is sufficient.”

The manuscript has been revised to better express the novelty. We thank the reviewer for this critical comment and apologize that the novelty and advantages of the current method were not adequately elaborated. The length scales accessed are necessarily limited, but this is true of any experiment, and we argue that the present measurements on the semi-microscopic length scale (crosslink segments) are of intrinsic interest precisely because they are larger than actin monomers but less than mesoscales probed by microrheology. Regarding time scale, it is true that this microscopy technique cannot measure more rapidly than ms, but this is true of any microscopy experiment, and we argue that the slow relaxation of these crosslinked gels renders this limitation not major.

The revised manuscript further clarifies the reasons to take the present approach of measuring crosslink fluctuations rather than more classical approaches. Indeed, various active microrheology (optical or magnetic tweezers) or AFM measurements can exert an external force on probes to effectively measure mechanical properties for materials of high modulus. But all these methods are fundamentally limited by their probe size (spherical particles or cantilever tips, submicron to microns). Their continuum-based theoretical frameworks cannot be extended to the molecular scale. Our methodology takes a step further and reaches the molecular scale. In particular, the proposed minimal model considers molecular junctions beyond the generalized Stokes-Einstein theorem that treats the network and solvent as a continuum, and the nontrivial fluorescent labeling specific to the crosslinks allows noninvasive tracking of them.

In microrheology, the probe particles must possess well-defined sizes a few times the network mesh size (on the order of $1\ \mu\text{m}$), precluding the use of molecular crosslinks or network segments as probes. The introduction of external particles is prone to be invasive as they may disturb network architecture by creating local “pockets”. On the other hand, there are many reports of single molecular tracking of

proteins and small molecules *in vivo* and *in vitro*. In those cases, the proteins or small molecules are free (or hindered to some extent) to diffuse but are not tethered to a network, so their dynamics merely reflects matrix viscosity. In our case, the crosslinks themselves are a part of the network, so their fluctuations directly reflect network elasticity. Respectfully, on these grounds we submit that the current methodology, presenting an alternative approach to the classical approaches, is significant as it offers information inaccessible otherwise. As we now better clarify, it cannot of course replace the important existing approaches, but it offers significant complementary information.

2. Concerns about the title.

“The title seems a bit misleading as the focus of the work is the proof of concept of the minimal model the authors developed (since the calculation of elasticity is based on it).”

The title has been revised to meet the reviewer’s constructive suggestion. Agreeing that this work focuses on calculating elasticity by fluctuations and on the subsequent mapping, and stressing the measurement of fluctuations on single molecular scale, the new title is “Single-crosslink microscopy in a biopolymer network to bridge molecular fluctuations and local elasticity”.

3. Concerns about crossover length.

“ensemble-averaged analysis of cross-correlation among different crosslinks as well as the network elasticity from a statistical perspective, rely on values averaged over the whole actin network. What is the critical/threshold length-scale that differentiates the measured elasticity in this study compared to other mesoscopic properties?”

The revised manuscript clarifies this important point with discussion and the addition of a new figure reproduced below (Fig. 4D in the main text). Briefly, we agree the analysis across length scales to identify possible crossover would be informative. Therefore, we derive elasticity on mesoscale length scales by dividing a full map into smaller tiles and averaging elasticities within each tile, as sketched in the figure below. The mesoscopic elasticity gradually converged to a common mean when the length scale approached the full size of our microscope view, 82 μm . It is noteworthy that if all the crosslinks (labeled and unlabeled) are counted, the convergence may take place at a smaller length scale, as we clarify in the revised text.

4. Concern about the biological relevance.

“the authors should briefly mention biological cytoskeletal networks that this biomimetic system tries to represent, for strengthening the significance of study.” “it would be better if the authors can illustrate the specific role of such a system and the underlying local crosslink mechanism in a biological context.”

The revised manuscript clarifies this important point. We thank the reviewer for this insightful suggestion and added the following discussion to the Introduction section. “We choose to implement the method and verify the model in an *in vitro* network of actin filaments for their significant biological relevance. Prevailing in eukaryotic cells, actin is the main constituent of the cytoskeleton—a network of filaments, crosslinking proteins, and motor proteins that plays a central role in cellular deformation, mobility, and division. Although elasticity of the cytoskeletal networks is not well understood on the molecular level, it is expected to be intimately related to force generation and transmission in living cells.” Meanwhile, possible implications of the present work in the biological context are now added to the Discussion section.

5. Concern about the possible roles of F-actin twisting and bending.

“Since it has been hypothesized that the elastic free energy of F-actin at small deformations is heavily affected by the twisting and bending motions due to its intrinsic structure, which emphasizes the significance of the transverse response as opposed to the entropic behaviors giving rise to the parallel response in the study.”

The revised manuscript discusses this point. As suggested by the reviewer and demonstrated for example in the paper by De La Cruz, Martiel, and coworkers. (Origin of Twist-Bend Coupling in Actin Filaments. Biophys. J. 2010), the bending, twisting, and their coupling indeed are important because of the double-stranded, helical organization of actin subunits; the twisting and coupling persistence lengths are respectively 1.4 and 0.4 μm , relevant to the crosslink distance on the order of 1 μm in this paper. The authors suggested that these modes are prominent on a time scale of μs to ms and at small filament deformations like subunit incorporation (~ 8 nm). In the current experiments, the frame rate is slower than this (0.5 s) and deformations are larger (~ 60 nm). We thus argue that bending, twisting, and their coupling do not significantly contribute to the crosslink fluctuations, and add this discussion to the Discussion section.

6. Concern about the assumption of cubic geometry.

“This seems problematic as there is no justification or reference from literature for this quantification. If the volume of a single F-actin is calculated in 3D, the deformation Delta should be calculated in 3D for consistency in terms of dimension. Does the deformation have specific directionality in a 3D vector form, or is it assumed to be simply 2D”

The revised manuscript fixes our earlier mistake and clarifies that the measurements are 2D. As we agree that directionality matters, the relevant calculations have been redone. In a classic paper on actin networks by Gardel, Weitz and coworkers (Elastic Behavior of Cross-Linked and Bundled Actin Networks, Science 2004), the authors considered the average filament between two crosslinks to occupy

an $\ell_c \times \xi \times \xi$ box, where ℓ_c is the distance between crosslinks and ξ the mesh size or distance between filaments. Following this argument, we have assumed the filament between crosslinks of a length ℓ_c to occupy an $\ell_c \times \xi \times \xi$ box in space. Directionality of the deformation is defined with respect to the filament. For deformation transverse to the filament (Δ_{\perp} , bending), the stress is $k_{\perp} \Delta_{\perp} / (\xi \ell_c)$ and the strain Δ_{\perp} / ξ and the elasticity is given by stress over strain, leading to $E_{\perp} = k_{\perp} / \ell_c = 3 \kappa / \ell_c^4$, identical to the equation we derived before. For deformation parallel to the filament (Δ_{\parallel}), the stress is $k_{\parallel} \Delta_{\parallel} / \xi^2$, the strain $\Delta_{\parallel} / \ell_c$, and the elasticity $E_{\parallel} = k_{\parallel} \ell_c / \xi^2 = 6 \kappa^2 / (k_B T \xi^2 \ell_c^3)$. This expression for E_{\parallel} is similar to Eq 2 in the paper by Gardel and Weitz (Science 2004) and we agree with the reviewer that the $\ell_c \times \xi \times \xi$ box assumption describes the current networks better than the cubic geometry does. Therefore, we have used this new equation in the revised manuscript and redone the calculations of E_{\parallel} (relevant to Figure 2G, spindle colors in Figure 3, and Figure 4C). No other calculations were affected.

7. Concern about possible origins of the negative correlations.

“I noticed that a negative correlation (Figure 3) only occurs between adjacent crosslinks with relatively different E_{parallel} , which might be worth discussing about.” “It seems counterintuitive that the network of interest in this study, which is only experiencing subtle thermal fluctuation, would result in a state far from equilibrium. As the authors have mentioned, potential buckling might result from crosslinks. However, statistics do not show bending-dominated responses. Therefore, it is possible that twisting might play a role.”

Additional discussion and a new figure have been added to the revised manuscript. We agree that non-thermal factors may not dominate. Twisting could indeed play a role but it is difficult to assess at present. We speculate regarding contributions from the anisotropic nature of actin filaments. Filament bending shortens or lengthens the distance between two crosslinks, causing possible anticorrelations (Supplementary Fig. 1A). Considering 4 crosslinks on rhombus corners (Supplementary Fig. 1B), the diagonal motion of one crosslink would cause two neighboring crosslinks to move against or away from each other. A new figure added to Supplementary Fig. 1 is reproduced below:

8. Concern about the proposed configuration of the actin-crosslink network in Figure 3.

“Are the filaments equally spaced by crosslinks? What is the average length assumed and measured in this work?”

As the revised manuscript now better clarifies, we did not assume specific configurations for the network except for the basic ones made in the minimal model. As only a handful of filaments are visible (similar to the image in Figure 1D), we cannot resolve the entire network. Since we have used the $\ell_c \times \xi \times \xi$ box assumption, the relevant length scales are $\ell_c \sim 1 \mu\text{m}$ and $\xi \sim 0.7 \mu\text{m}$, which indeed is comparable to the measured values.

9. Concern about quantitative maps, cell geometry, and network percolation.

“the authors should provide quantitative maps in addition to the qualitative spatial map” “Is the heterogeneity in local elasticity dependent on cell geometry or network mechanical percolation?”

The revised manuscript now does a better job of expressing that the map in Figure 3 is considered to be quantitative because the fluctuation, elasticity, and correlation were all calculated from experimental data and their representations (spindle colors, sizes, line widths) all were scaled to the actual values. But as we now better clarify, the size of this map is limited by the microscope’s field of view, and also as the image is in the 2D plane, we do not resolve filaments that might be vertical or oblique to the focal plane. In the experiments, we used a cell with $120 \mu\text{m}$ depth and imaged the filaments $30 \mu\text{m}$ away from the glass wall to minimize any wall effect. As long as the cell size is much larger than the filament length, we expect no strong dependence of heterogeneity on the cell geometry; there could be heterogeneity approaching the cell wall as a few reports in the literature have suggested, but we did not check for this, preferring to simply avoid proximity to the wall. As for network percolation and its dependence on crosslink density, we agree that one might anticipate this on physical grounds, but to do so is beyond the scope of the present experiments. These points are now discussed in the revised manuscript.

10. Concern about the influence of filament lengths and other cytoskeletal-related properties.

“In similar actin networks, filament length has been shown to be a crucial player in regulating elasticity.” “It would be clearer if the authors clarify the length scales of filaments both in vitro and in silico in this study.” “how sensitive would the proposed network model be as a response to other cytoskeletal-related properties?”

The revised manuscript discusses these issues and we thank the reviewer for this critical concern. As suggested by the reviewer and shown in the paper by Kasza, Weitz and coworkers (Actin filament length tunes elasticity of flexibly cross-linked actin networks. Biophys. J. 2010), the filament length indeed plays an important role in determining the network elasticity both for flexible crosslinks (filamin A) and rigid crosslinks (biotin-avidin) in contrast to the prediction of classical affine theory. The former is attributed to the compliant nature of filamin A crosslinks and the latter to nonaffinity of the network. In this paper, we intended to keep our model minimal and therefore chose a network condition (rigid junction of biotin-avidin and filament length \gg crosslink distance) to minimize the number of variables; we took the approach that it’s best to begin with a minimal model that later can be

elaborated as needed. Flexible and/or dynamic crosslinking is certainly an element worth considering, since many biological crosslinkers are of flexible and dynamic nature, and other factors may include filament length, filament treadmilling, and crosslinker functionality, for example. Theoretical and experimental treatment of these factors are beyond the scope of this paper, as we now do a better job of clarifying. Relevant discussion and references have been added to the revised manuscript.

11. Concern about the crosslink rigidity.

“Uncrosslinked inter-filament networks might be softened more easily. The authors attributed this to dominance of structural forces in uncrosslinked actin systems. In crosslinked networks, I wonder if stiffening of crosslinks would affect network elasticity in this presented system.”

Now discussed in the revised manuscript. Indeed, uncrosslinked but entangled actin networks are softer than the crosslinked counterpart in this work. Intuitively we agree with the referee’s implication that softening of crosslinks should affect network elasticity in the present system but quantitative evaluation of this is beyond the scope of this paper.

12. Concern about external forces and myosin motors.

“if the authors would extend the methodology to incorporate external forces, how would myosin motors be configured? How would the length-scale be manipulated?”

The revised manuscript clarifies the importance of these other issues. Indeed, we are working on developing a microscope-compatible rheometer. The strain-stiffening common to many biological networks should reduce thermal fluctuations. Myosin action on two neighboring filaments should contract them and we speculate that addition of myosin to the current, rigidly crosslinked network may effectively decrease l_c and actively move the crosslinks beyond the thermal fluctuations. The revised manuscript includes this discussion.

13. Concern about the slight but systematic offset in Figure 2C.

The revised manuscript clarifies that indeed there is a slight, systematic overestimation in Figure 2C, which may suggest secondary contributions from unequal l_c or off-orthogonal configurations that are not incorporated into the minimal model.

14. Typo in the caption of Figure 4.

Fixed.

15. Concern about trajectory ellipticity in Figure 1E.

The revised manuscript clarifies that typical trajectories are indeed not necessarily elliptical, an observation in line with our proposed spindle trajectory in 3D. As the orientations of spindles are random, their projections on the focal plane can be circular ($\theta = 90^\circ$ in the figure below). We presented

typical trajectories in Fig. 1E to show that the ellipticity in 2D is random, and the trajectory area is polydisperse. This point is now clarified in the revised manuscript and a new panel added to Fig. 2C is reproduced below:

Response to Reviewer # 2

The reviewer recommended publication based on a perception that the new perspective provided by the current methodology may also have larger potential applications in other soft materials systems. All concerns were met:

1. Question about the spatial resolution.

“One minor comment: What is the spatial resolution for this single-crosslink microscopy?”

The revised manuscript clarifies that the minimal resolvable distance between two crosslinks is limited by optical resolution ~ 200 nm in the current setup, while the minimal detectable fluctuation of the crosslinks is limited by the stability and noise of the optical setup, ~ 20 nm. In principle, the former resolution could be improved to ~ 40 nm using super-resolution microscopy, and the latter to ~ 10 nm if the system stability and noise were further improved, but all of this would be hard work. Thanks for helping us to clarify this important point.

Response to Reviewer # 3

We thank the reviewer for the positive feedback and critical and insightful comments. We have done our best, in the revised manuscript, to address the reviewer's concerns.

1.1 Concerns about insufficient explanation.

“The authors need to clarify in several cases to explain their methods and calculations at an appropriate level and to justify their conclusions. Substantial revisions are needed and it may be that the manuscript has to be significantly lengthened to provide adequate explanation.”

The revised manuscript is significantly longer with a better background introduction, more details about the theoretical model and experiments, and more in-depth discussion. In particular, the main text is lengthened from 1800 to 3200 words and 5 new figure panels are added. The novelty of this methodology and its differences from the classical microrheology approach are emphasized in the Introduction section. The Results section is divided into several subsections that address reviewers' comments. The Discussion section has been extended to discuss several open questions raised by the reviewers, including the relevant length scales, bending and twisting modes, anticorrelations, external/internal strains, and future prospects.

1.2 Question about the concentration of labeled actin filaments.

“What does “trace” mean? This is important information because it sets the fraction of crosslinks that are visible by fluorescence.”

As we now clarify, the unlabeled/labeled filament ratio is ~1000:1 and the roughly half-half ratio in Fig 1C is only for illustrative purpose.

1.3 Concerns why labeled filaments are visually parallel to the focal plane.

“there would appear to be no reason why the labeled filaments should have a special orientation, does this mean that there is a measurement bias, in which filaments normal to the plane or not detected?”

The revised manuscript does a better job of explaining that while labeled filaments should indeed be randomly orientated, experimentally we tracked labeled filaments that lie in the focal plane as these we could best resolve, and we disregarded filaments normal to the plane.

1.4 Concerns about the connections and distances between labeled and unlabeled crosslinks.

“it was not clear how the authors know reliably which of the visible crosslinks are connected by filaments and which are not.” “how do the authors know that the distance between two visible crosslinks is the distance between actual neighboring crosslinks?”

Clarified in the revised manuscript. The unlabeled crosslinks overwhelm the labeled ones (1000 times more abundant), so it is indeed much more likely for the labeled ones to be connected to unlabeled ones. As we now state more precisely, the connections drawn in Figure 3 do not suggest direct connections between labeled crosslinks. In this paper, l_c represents distance between any pair of

neighboring crosslinks regardless of whether they were imaged; we did not measure the distances between labeled crosslinks nor assume such distances as l_c , as the reviewer is of course correct that we cannot directly measure l_c as most crosslinks are not labeled. Instead, it is calculated for each measured fluctuation magnitude b . Specifically, we have $b^2 = 1/(3 \ell_p/\ell_c^3 + 6 \ell_p^2/\ell_c^4)$ from Eqs 7 and 11. Given $\ell_p = 16 \mu\text{m}$ for actin filaments, we can get a l_c for each measured b . A calculation workflow is now added to Fig. 2E (reproduced below). These two points are now made clear in the Methods section.

2. Concerns about the spindle orientation and size in Figure 3.

“How do the authors know the 3D orientations and sizes of the ellipsoids shown in Fig 3?”

Clarified in the revised manuscript. In the Methods section, the orientation and size of a fluctuation ellipsoid is defined by θ (Fig. 2C) and Δ_{\perp} and Δ_{\parallel} (thermal fluctuation magnitudes vertical and parallel to the filament plane). These quantities are related to 2D measurables by $a = \Delta_{\perp} \cos \theta$ (Eq. 10) and $b = \Delta_{\parallel}$ (Eq 11), where a and b are the long and short axis lengths of the 2D elliptical trajectory. In practice, we measure b to calculate Δ_{\parallel} (Eq 11) and Δ_{\perp} (Eqs 6 and 7, given $l_p = 16 \mu\text{m}$), so the spindle size is fully determined. From a and Δ_{\perp} , θ is then determined. This detailed procedure is now presented in Fig. 2E.

3. Concerns about how the map (Figure 3) was constructed.

“are these crosslinks all in the same image plane? From my understanding of the procedure, the correlations can only be measured for pairs of crosslinks that are in the same plane. However, the illustration conveys a sense of perspective, as if the correlations were measured for crosslinks that were in different planes. How as that done? Also, what is the scale for this image?”

As clarified in the revised manuscript, these crosslinks are in the same focal plane with a thickness $\sim 1 \mu\text{m}$. Although we generated the image using 3D drawing software (3DS max) to illustrate different orientations of the different spindles, the crosslinks are in the same plane, so their correlations are likewise calculated in this plane. The revised manuscript specifies the image size, $42 \times 45 \mu\text{m}$.

4. Concerns about the theoretical curve of a' and b and about Lc.

“What is the equation that represents the theory curve of Fig 2C?” “I very roughly estimated the ratio of L_c/L_p and found a value of L_c that was on the order of a few microns. This seems much too large for L_c if it is to represent the distance between actual (not just visible) crosslinks.”

Clarified in the revised manuscript. In experiments we collected a large number of trajectories, determined their a and b , then binned the a values according to b values, $a' = \langle a \rangle_b$. In our model, 3D spindles with a given Δ_{\parallel} take random orientations θ so their projections in 2D are statistically averaged by θ , giving $a' = \langle \Delta_{\perp} \cos \theta \rangle_{\theta}$ independent of a particular θ . The equality of experimentally measured a' and theoretically predicted a' is confirmed below (grey points and grey line, $\ell_p = 16 \mu\text{m}$ is an input). Compared to the previous version, the figure reproduced below (Fig. 2F) contains new predictions regarding ℓ_p , Δ_{\parallel} , and Δ_{\perp} .

“how do these compare to what one might expect from the known concentration of crosslinkers?”

In the revised manuscript, we summarize the length scales relevant to this work in a new diagram (Fig. 2H, reproduced below). Known lengths specific to actin filaments include contour length $\ell \sim 10 \mu\text{m}$, persistence length $\ell_p = 16 \mu\text{m}$, and monomer size = 8 nm. The mesh size or average distance between filaments ξ can be estimated from actin concentration c in mg/ml by $\xi \approx 0.3/\sqrt{c}$ in μm , giving $\xi \approx 0.7 \mu\text{m}$ for 0.2 mg/ml. Length of green labeled segments ~ 100 nm defines the probe length scale of the current single-crosslink method, an order of magnitude smaller than that of microrheology ($\sim 1 \mu\text{m}$). The measured fluctuation magnitude b ranges from 30 nm to 200 nm (Fig. 2H, cyan area, and Fig. 4A), corresponding to crosslink distance ℓ_c from $0.7 \mu\text{m}$ to $2.5 \mu\text{m}$ (Fig. 2H, pink area, and Fig. 4B). Considering that each crosslink consumes two biotin sites and assuming all the biotin sites are crosslinked, we estimate the lower limits of average ℓ_c to be 1.5, 0.6, and $0.3 \mu\text{m}$, respectively, for biotin concentrations = 0.5, 1.5, and 3 % and a given monomer size = 8 nm. In comparison, the experimental peaks ℓ_c (Fig. 4B) are 1.8, 1.4, and $1.1 \mu\text{m}$, respectively, reasonably larger than the lower limits; the discrepancy increases probably because relatively more biotin sites are not crosslinked at higher biotin concentration.

5. Concerns about negative correlations.

“thermal excitations cause this filament to bend, something like a guitar string being plucked. In such a case, the two end points would move toward one another.” “The other sign of correlation would correspond to the two imaged spots tending to move in the same direction, which could be from long-wavelength modes of vibration.”

The revised manuscript discusses this. We thank the reviewer for the thoughtful suggestions and agree that non-thermal factors may not dominate in this system. As we now speculate, contributions from the anisotropic nature of actin filaments may contribute; filament bending shortens the distance between two crosslinks, causing possible anticorrelations (Supplementary Fig. 1A). Considering 4 crosslinks on rhombus corners (Supplementary Fig. 1B), the diagonal motion of one crosslink could cause two neighboring crosslinks to move against or away from each other. This discussion and the figure reproduced below are added to the revised manuscript as Supplementary Fig. 1.

6. Concerns about the section titles.

We reorganized the paper substantially with new subsection titles. Thanks.

7. Concerns about the phrase “thermal energy acts on the free end”.

Omitted in the revised manuscript. Thanks.

Reviewers' comments:

Reviewer #1 (Remarks to the Author):

In the revised manuscript, the authors provided sufficient data and results to address my major concerns. Point-by-point responses are included in the main text, and the discussion section has been improved significantly. The justification of the impact and novelty seems adequate. So, I recommend publication.

Reviewer #2 (Remarks to the Author):

My minor comment on “the special resolution for this single-crosslink microscopy: has been satisfactorily addressed. I also looked over the comments from the other two reviewers, and felt their comments were addressed by the authors too. Therefore, I strongly support the publication of this work in its current form.

Below are my thoughts on the comments from the two reviewers (**Reviewer #1** and **Reviewer #3**) and the replies from the authors:

Reviewer #1

1. 1. Concerns about novelty.

“But, in the mechanical perspective, it would be much more efficient to use an AFM-based system or an optical tweezer system to directly quantify the deformation level and time evolutions of forces and velocities of single proteins or particles. The authors should address the novelty and advantages of the method and analysis in an experimental standpoint more clearly.”

“Results from the model seem scientifically sound, but it also requires further validation to improve its versatility given the limited spatiotemporal scale and dynamic state. Therefore, I doubt the impact of the work in its current form is sufficient.”

My thought: The authors have successfully demonstrated the novelty of the paper. In particular, current methodology reaches the molecular scale, which is not achievable by other continuum-based theoretical frameworks such as AFM. The crosslinks themselves are a part of the network, therefore their fluctuations directly reflect network elasticity, rather than the matrix viscosity, which is measured by other methods.

1.2&1.3 Concerns about the title and crossover length

My thought: The authors replied appropriately.

1.4. Concern about the biological relevance.

“the authors should briefly mention biological cytoskeletal networks that this biomimetic system tries to represent, for strengthening the significance of study.” “it would be better if the authors can illustrate the specific role of such a system and the underlying local crosslink mechanism in a biological context.”

My thought: Possible implications of the present work in the biological context have been now added to the Discussion section by the authors.

1.5 Concern about the possible roles of F-actin twisting and bending.

“Since it has been hypothesized that the elastic free energy of F-actin at small deformations is heavily affected by the twisting and bending motions due to its intrinsic structure, which emphasizes the significance of the transverse response as opposed to the entropic behaviors giving rise to the parallel response in the study.”

My thought: Comment is well addressed.

1.6 Concern about the assumption of cubic geometry.

“This seems problematic as there is no justification or reference from literature for this quantification. If the volume of a single F-actin is calculated in 3D, the deformation Δ should be calculated in 3D for consistency in terms of dimension. Does the deformation have specific directionality in a 3D vector form, or is it assumed to be simply 2D.”

My thought: In the revised manuscript, the authors fixed the mistake and clarifies that the measurements are 2D.

1.7. Concern about possible origins of the negative correlations.

“I noticed that a negative correlation (Figure 3) only occurs between adjacent crosslinks with relatively different E_{parallel} , which might be worth discussing about.” “It seems counterintuitive that the network of interest in this study, which is only experiencing subtle thermal fluctuation, would result in a state far from equilibrium. As the authors have mentioned, potential buckling might result from crosslinks. However, statistics do not show bending-dominated responses. Therefore, it is possible that twisting might play a role.”

My thought: The authors added new experimental data to address this concern. A new figure has also been added.

1.8& 1.9 Concern about the proposed configuration of the actin-crosslink network in Figure 3. and Concern about quantitative maps, cell geometry, and network percolation

My thought: The authors have modified and clarified the misleading content in the paper.

1.10 & 1.11 Concern about the influence of filament lengths and other cytoskeletal-related properties. & Concern about the crosslink rigidity

“In similar actin networks, filament length has been shown to be a crucial player in regulating elasticity.” “It would be clearer if the authors clarify the length scales of filaments both in vitro and in silico in this study.” “how sensitive would the proposed network model be as a response to other cytoskeletal-related properties?” “Uncrosslinked inter-filament networks might be softened more easily. The authors attributed this to dominance of structural forces in uncrosslinked actin systems. In crosslinked networks, I wonder if stiffening of crosslinks would affect network elasticity in this presented system”

My thought: Reviewer 1 has raised very good points in my view. As the author correctly mentioned, they admitted the influence of filament lengths and the crosslink rigidity. However, at current stage, they are unable to explore the effects of these factors.

1.12~1.15 Minor comments

My thought: Minor comments have been fixed accordingly.

Reviewer #3

3.1 Concerns about insufficient explanation, the concentration of labeled actin filaments, why labeled filaments are visually parallel to the focal plane and the connections and distances between labeled and unlabeled crosslinks.

3.2 Concerns about the spindle orientation and size in Figure 3. “How do the authors know the 3D orientations and sizes of the ellipsoids shown in Fig 3?”

3.3 Concerns about how the map (Figure 3) was constructed.

“are these crosslinks all in the same image plane? From my understanding of the procedure, the correlations can only be measured for pairs of crosslinks that are in the same plane. However, the illustration conveys a sense of perspective, as if the correlations were measured for crosslinks that were in different planes. How is that done? Also, what is the scale for this image?”

My thought on the authors’ replies on 3.1-3.3: It is indeed difficult for the readers to follow the authors’ ideas unless reading back and forth with extreme attention. To this end, the reviewer raised very good comments. In the revised manuscript, the authors have added detailed explanation with necessary figures. The misleading statements have also been clarified. The revised manuscript has greatly improved in terms of clarity.

3.4 Concerns about the theoretical curve of a' and b and about L_c .

“What is the equation that represents the theory curve of Fig 2C?” “I very roughly estimated the ratio of L_c/L_p and found a value of L_c that was on the order of a few microns. This seems much too large for L_c if it is to represent the distance between actual (not just visible) crosslinks.”

My thought: The authors attributed such large distance to that relatively more biotin sites are not crosslinked at higher biotin concentration. Even though the comment is not addressed perfectly, it is the best that the authors can reply.

3.5 Concerns about negative correlations.

“thermal excitations cause this filament to bend, something like a guitar string being plucked. In such a case, the two end points would move toward one another.” “The other sign of correlation would correspond to the two imaged spots tending to move in the same direction, which could be from longwavelength modes of vibration.”

My thought: The comment has been well addressed by the authors. Non-thermal factors may not dominate in this system and the diagonal motion of one crosslink could cause two neighboring crosslinks to move away from each other, which, as a result, causes the negative correlations.

3.6 & 3.7 Minor comments

My thought: Minor changes have been made accordingly.

Reviewer #3 (Remarks to the Author):

The authors have addressed all of my major comments. The additional text helps a great deal to clarify what was done. I do, though, have some concerns remaining.

1) P16, L125 and P13,L12: I still have a question: I still do not understand what the "theory" is relating a' and b' . The authors have already used the relations (6,7) and (11,12). I don't see that there is a truly independent test to be made of these relations, except perhaps to histogram the extracted theta values and see if the blobs are randomly oriented. My confusion can perhaps be clarified if the authors would state explicitly what the procedure was for carrying out their calculation to obtain Fig 4F,G. Is it just a rearrangement of an equation given in the text? Is it a simulation? If they used equation(s) in the text, then does this really provide an independent test of the assumptions used in the analysis?

2) a simple recommendation about Fig 3: I understood the authors' response to my question about computing the correlation between crosslinks: the crosslinks do indeed have to be in the same plane for this calculation to be done. In this case, I still find Fig 3 to be very confusing: it looks 3D but it isn't. I urge the authors to write this fact explicitly in the caption to help inform the reader. In addition, perhaps the correlation bars should be shown without shading?

3) A minor comment: The caption of Fig S2 is needlessly hard to follow. It mixes procedure, data presentation, and interpretation in an awkward way. Typically, a figure caption would first say what the axes are and what the symbols mean (in other words, what was actually done), and then describe the trends and conclusion. In this case, the authors have chosen to put the conclusion first and include the definitions of what the symbols mean in parentheses. It needlessly takes time to unpack captions like this. If possible, I suggest that this be changed. This is a minor comment but I mention it because I still find this a challenging paper to parse for readers who want to know how to repeat this work. Therefore, anything that the authors can do to clarify the facts about data and procedure will help.

Revision Requested for Manuscript NCOMMS-19-01878A

Single-crosslink microscopy in a biopolymer network dissects local elasticity from molecular fluctuations

Response to Reviewers # 1 & # 2

Both reviewers were satisfied with the revised manuscript and recommended publication in its present form. We thank them.

Response to Reviewer # 3

The reviewer stated that the revised manuscript “addressed all of my major comments.” Regarding the additional suggestions:

1. Questions about the calculations. The manuscript has been revised to clarify these points. Specifically:

As we have related the key parameters to the two measurables by eq. 6, 7, 11, and 12 (Fig. 2E), we proceed to calculate their dependence on b using the input parameter of $\ell_p = 16 \mu\text{m}$ for actin filaments (Fig. 2F). Rearrangement of eq. 7 and 12 gives $b^2 = 1/(3 \ell_p/\ell_c^3 + 6 \ell_p^2/\ell_c^4)$, which is numerically solved to predict the dependence of ℓ_c on b (black line). This dependence, along with eq. 6, is further used to calculate the dependence of Δ_{\perp} on b (blue line). Dependence of Δ_{\parallel} on b is given by eq. 12 (red line). In our model, 3D spindles with a given Δ_{\parallel} take random orientations θ so we define an averaged quantity a' independent of a particular θ , given by $a' = \langle a \rangle_{\theta} = \langle \Delta_{\perp} \cos \theta \rangle_{\theta}$. Dependence of a' on b is also calculated numerically (grey line). Experimentally, 2D trajectories of similar b values can vary significantly in a because their 3D orientations (θ) are random. We thus group all the measured trajectories into subsets according to their b values and obtain an averaged a value for each subset (a' , grey points). Reasonable consistence between the measured and predicted a' values (grey points and grey line) therefore validates our minimal model.

Please note that as the measurements of a' and b require no pre-assumptions nor involve any equations we derived, they do provide an independent test of our model. For example, if we were to assume $\ell_p = 1 \mu\text{m}$, the model would predict $a' \approx b$, but our measured a' and b values would remain unchanged and thus disagree with this input value.

In Fig. 2G, the two input values are $\ell_p = 16 \mu\text{m}$ and mesh size $\xi = 0.7 \mu\text{m}$ for 0.2 mg/ml. For each b value, ℓ_c is calculated as above [$b^2 = 1/(3 \ell_p/\ell_c^3 + 6 \ell_p^2/\ell_c^4)$]. Then E_{\perp} is calculated by eq. 13 and E_{\parallel} by eq. 14. These two predicted lines are not compared to experimental measurements as we discussed in the second paragraph in General aspects in Discussion section.

Lines in Fig 4F and 4G are simple exponential and power-low fittings for crosslink pairs, irrelevant to the minimal model for single crosslinks. The data points are measured per definition of cross-correlation (eq. 15).

In the revised manuscript, the main text and figure captions are modified to clarify these points. We have also slightly changed the title into “Single-crosslink microscopy in a biopolymer network

dissects local elasticity from molecular fluctuations” to stress the dependence of fluctuations on local elasticity.

2. Recommendation regarding the presentation of Fig 3.

The revised manuscript now clarifies, in the figure caption, that Fig. 3 is 2D. The shading of the correlation bars is removed, and a dashed grid is added to stress the 2D nature of the map.

3. Concerns about the clarity of the caption of Fig S2.

The revised manuscript follows the reviewer’s suggestion; the caption is rearranged as recommended.

REVIEWERS' COMMENTS:

Reviewer #3 (Remarks to the Author):

The authors have addressed all of my comments. I recommend publication for the reasons that have been described earlier.